# Prediction and verification of the AD-FTLD common pathomechanism based on dynamic molecular network analysis

Meihua Jin[1,6], Xiaocen Jin[1,6], Hidenori Homma[1,6✉], Kyota Fujita[1], Hikari Tanaka[1], Shigeo Murayama[2,3], Hiroyasu Akatsu[4], Kazuhiko Tagawa[1] & Hitoshi Okazawa[1,5✉]

Multiple gene mutations cause familial frontotemporal lobar degeneration (FTLD) while no single gene mutations exists in sporadic FTLD. Various proteins aggregate in variable regions of the brain, leading to multiple pathological and clinical prototypes. The heterogeneity of FTLD could be one of the reasons preventing development of disease-modifying therapy. We newly develop a mathematical method to analyze chronological changes of PPI networks with sequential big data from comprehensive phosphoproteome of four FTLD knock-in (KI) mouse models (PGRN$^{R504X}$-KI, TDP43$^{N267S}$-KI, VCP$^{T262A}$-KI and CHMP2B$^{Q165X}$-KI mice) together with four transgenic mouse models of Alzheimer's disease (AD) and with APP$^{KM670/671NL}$-KI mice at multiple time points. The new method reveals the common core pathological network across FTLD and AD, which is shared by mouse models and human postmortem brains. Based on the prediction, we performed therapeutic intervention of the FTLD models, and confirmed amelioration of pathologies and symptoms of four FTLD mouse models by interruption of the core molecule HMGB1, verifying the new mathematical method to predict dynamic molecular networks.

[1] Department of Neuropathology, Medical Research Institute, Tokyo Medical and Dental University, Bunkyo-ku, Tokyo, Japan. [2] Department of Neuropathology, Brain Bank for Aging Research, Tokyo Metropolitan Institute of Gerontology, Itabashi-ku, Tokyo, Japan. [3] Brain Bank for Neurodevelopmental, Neurological and Psychiatric Disorders, Molecular Research Center for Children's Mental Development, United Graduate School of Child Development, Osaka University, Suita, Osaka, Japan. [4] Department of Medicine for Aging in Place and Community-Based Medical Education, Nagoya City University Graduate School of Medical Sciences, Nagoya, Aichi, Japan. [5] Center for Brain Integration Research, Tokyo Medical and Dental University, Bunkyo-ku, Tokyo, Japan. [6] These authors contributed equally: Meihua Jin, Xiaocen Jin, and Hidenori Homma. ✉email: homma@m.ieice.org; okazawa-tky@umin.ac.jp

Molecular network analysis in biology is a powerful tool to unravel the roles of previously unknown protein–protein interactions in physiological and pathological situations[1,2]. However, the application to the research of neurodegenerative diseases has been limited to a few cases[3–6], and such research has not been performed to find a common pathology across different types of neurodegenerative diseases. Another issue is that previous molecular network analyses have been static, i.e., performed only at a single time point. The method and theory lack explanation for a dynamic network that shifts chronologically. Employing frontotemporal lobar degeneration (FTLD) and Alzheimer's disease (AD) as objects, we investigated a dynamic molecular network that is common across multiple dementias.

FTLD is one of the most frequent causes of neurodegenerative dementia, which frequently associates with motor neuron disease[7–11]. A number of genes such as TAR DNA-binding protein 43 (TDP43)[12–14], progranulin (PGRN)[15,16], valosin-containing protein (VCP)[17,18], and charged multivesicular body protein 2B (CHMP2B)[19,20] are known causative for familial FTLDs. Hence, FTLD and related diseases have been referred to since recently as FTLD spectrum[21]. Protein aggregation of Tau, TDP43, or p62 is a pathological hallmark of FTLD and simultaneously a critical process mediating the pathological progression, leading to neuronal dysfunction and cell death[22,23].

However, liquid–liquid phase transition of causative proteins and of associating intrinsically disordered proteins (IDPs) indicates that such proteins in soluble state reversibly assemble and disassemble in inclusion bodies in neuronal cytoplasm and nucleus[24–30]. From the aspect of functions, disease proteins are involved in intracellular vesicular trafficking[31,32], RNA metabolism[33], DNA damage repair[34–36], and so on. Loss of physiological functions of disease-causative proteins owing to mutations is another but presumably complementary scheme to gain of functions that induce neuronal dysfunction and cell death[37,38].

Interestingly, we recently revealed that newly generated knock-in mouse models carrying a gene mutation of TDP43, PGRN, VCP, or CHMP2B at the conserved amino acid between mouse and human suffer DNA damage in embryonic neural stem cells owing to the loss of their DNA damage repair functions[39]. The developmental DNA damage is carried over to neurons differentiated from neural stem cells and induces early-stage necrosis[39] in FTLD mouse models and human patients like in the cases of AD model mice and human patients[40]. It is known that damage-associated molecular patterns/alarmins such as high mobility group box 1 (HMGB1) induce protein kinase C (PKC) signals via Toll-like receptors (TLRs), and the story applies to neurons surrounding necrosis in AD pathology[41]. Moreover, interruption of HMGB1-TLR4 axis by anti-HMGB1 antibody ameliorates pathological progression and symptoms of AD model mice[41].

The similarities of AD and FTLD pathologies in DNA damage and repair or in HMGB1-TLR-PKC pathway prompted us to investigate more comprehensively whether and how common pathomechanisms are shared between the two largest groups of dementia affecting the function and cell death of cerebral neurons together with intracellular[7–11,40,42,43] and/or extracellular[44] protein aggregations.

For this purpose, we developed in this study a new mathematical method for dynamic molecular network analysis of big data of comprehensive phosphoproteome from mass analyses of four FTLD mouse models (four genotypes of KI mice) and four AD mouse models (four genotypes of transgenic mice) at multiple time points during aging, and identified core common pathological network that changes dynamically. APP$^{KM670/671NL}$-KI mice[45] were also used for generating external hold-out data to confirm the validity of methods and results. We employed mouse models instead of human postmortem brain, human iPS cell model, or non-human primate model because we could compare the values with controls on the same genetic background, because we could continuously follow the chronological progression of pathology from early-stage to the terminal stage in vivo, and because preparation of multiple samples could be allowed ethically, which are difficult with human postmortem brains, human iPS cells, or non-human primates. We used the whole cerebral cortex as the object of mass analyses because AD and FTLD pathologies both affect the cerebral cortex and because divided regions of the cerebral cortex such as frontal or occipital cortex might produce certain variances in the results by dissection techniques. Moreover, we used human postmortem brains to verify that the results of mice were translatable to human AD and FTLD pathologies.

Surprisingly, the core AD-FTLD network, which connects PKC and relevant signals downstream of HMGB1-TLR4 to synapse function, was shared across multiple FTLD and AD pathologies. Moreover, we confirmed a therapeutic effect of a human anti-HMGB1 human antibody, which interrupts the predicted AD-FTLD common pathway by HMGB1-induced signals (Tanaka, H. et al. HMGB1 signaling phosphorylates Ku70 and impairs DNA damage repair in Alzheimer's disease pathology. unpublished), on the symptoms and pathologies of multiple FTLD mouse models, supporting the soundness and availability of our new mathematical method, together with verification by multiple mathematical confirmations.

## Results

**Mathematical analysis of sequential phosphoproteome big data.** We generated four FTLD mouse models, PGRN$^{R504X}$-KI, TDP43$^{N267S}$-KI, VCP$^{T262A}$-KI, and CHMP2B$^{Q165X}$-KI mice, as described previously[6,39], performed comprehensive and quantitative phosphoproteome analysis with total cerebral cortex tissues at 1, 3, and 6 months of age together with four types of AD mouse models (5xFAD, APP-Tg, PS1-Tg, PS2-Tg), obtained ratios of peptide signals between disease model and the control (Supplementary Figure 1a), converted the ratios to log values (Supplementary Figure 1b), confirmed their normal distribution (Supplementary Figure 1b, c), and selected changed phosphopeptides between FTLD model mice and non-transgenic sibling mice or between AD model mice and non-transgenic sibling mice when their p values were <0.05 in Welch's test ($N = 3$ mice) or q values <0.05 by post hoc Benjamini–Hochberg (BH) procedure for multiple hypothesis testing (Fig. 1, Step 1). Based on the integrated protein–protein interaction (PPI) database, we generated 24 pathological protein networks in a total of each mouse model at each time point (Fig. 1, Step 2; Supplementary Figure 2a, b). We compared proteins including significantly changed phosphopeptide(s) detected by mass spec analysis in more than two models (Supplementary Figure 3a–c), from which we selected core nodes possessing ordinary or high betweenness scores (>0 or >2 SD) reflecting their impact sizes on the pathological protein network (Fig. 1, Step 3).

In parallel, we selected edges to connect closely related nodes (Fig. 1, Step 4). In this case, we generated three-dimension (3D) vectors by using ratios of disease/control values at 1, 3, 6 months as x, y, z positions, and connected two nodes by a core edge when 3D vectors of the two nodes have a high correlation (correlation value ≥0.9) (Fig. 1, Step 5). The correlation analysis enabled us to select a combination of closely related nodes that revealed similar patterns of chronological changes. Core nodes and core edges generated core networks of FTLD and AD. Finally, we compared AD core network and FTLD core network

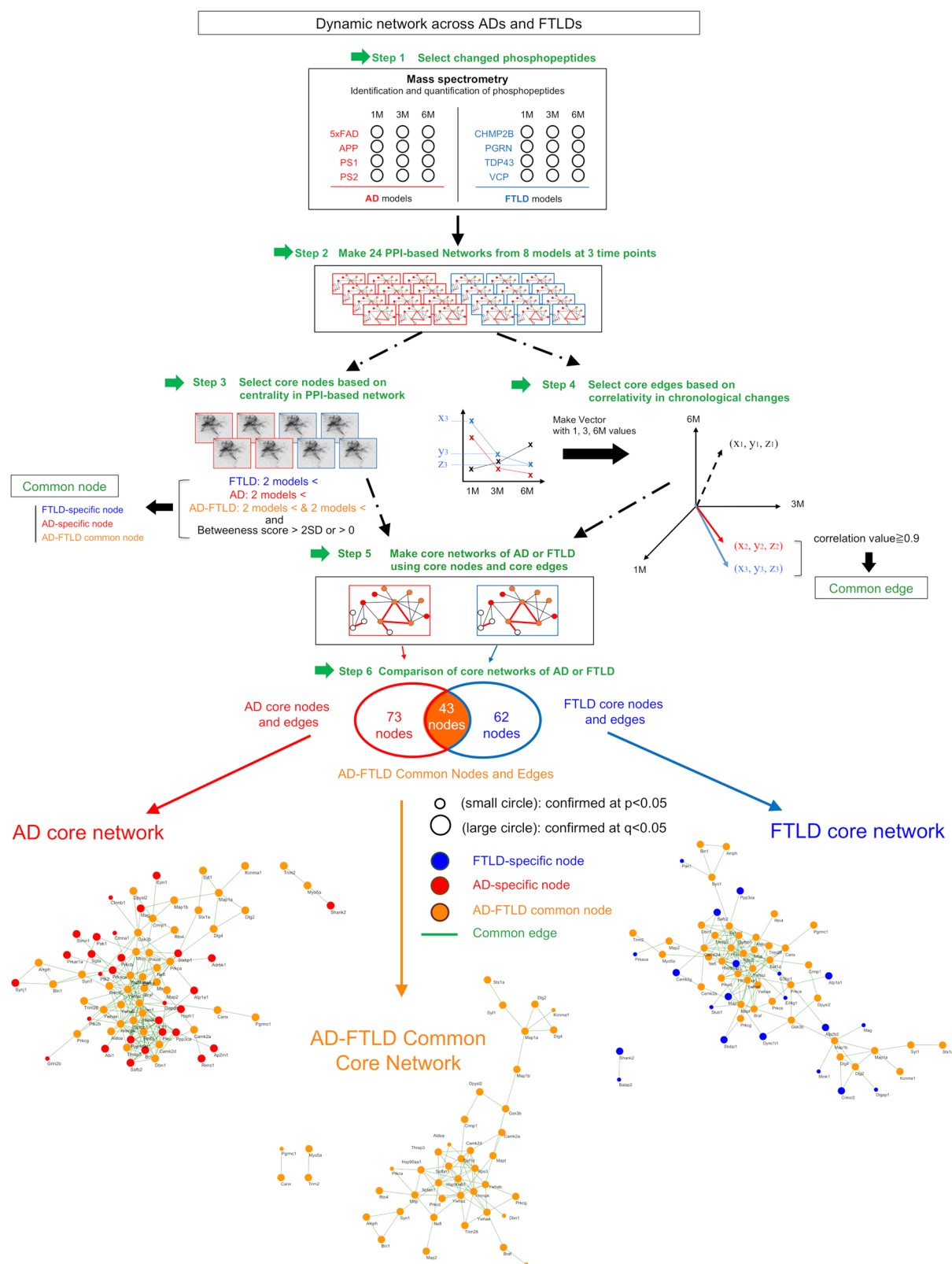

(Fig. 1, Step 6). Surprisingly, FTLD core network and AD core network were highly homologous and shared 46.7% of core nodes (Fig. 1). In addition, almost all the core nodes were selected not only by $p$ values (small circle) but also by $q$ values (large circles) (Fig. 1).

Moreover, we examined by permutation test how rarely we could expect 46.7% of commonness. We detected 1965 phosphoproteins in total, irrespective of their changes in disease condition through our phosphoproteome analyses of eight mouse models ($N = 3$) at three time points. By bootstrap procedure, we randomly selected 73 nodes (equal to the number of AD core nodes) and 62 nodes (equal to the number of FTLD core nodes), repeated the selection 10,000 times, and tested their overlap. The result of calculations revealed that %commonness revealed a

**Fig. 1 Flow chart of identification of dynamic molecular network shared across ADs and FTLDs.** Step 1: significantly changed phosphorylation sites were selected from comprehensive phosphoproteome analysis with total cerebral cortex tissues of four AD and four FTLD mouse models at 1, 3, and 6 months of age. Step 2: pathological protein network of each mouse model at each time point was generated based on PPI database as described in the methods. Step 3: protein nodes changed in more than two models and possessing a high score (>3 SD) of betweenness were designated as core nodes. Step 4: edges to connect closely related nodes were selected by correlation analysis of three 3D vectors in which ratios of disease/control values at 1, 3, 6 months were used as x, y, z positions. We defined core edge when two 3D vectors of two nodes at the end had a high correlation (correlation value ≥ 0.9). Step 5: core network was made based on core nodes and core edges. Step 6: core networks of AD and FTLD were compared to generate AD core network, FTLD core network, and AD-FTLD common core network. Small circles were selected only by $p$ values ($p < 0.05$ in Welch's test) while large circles were selected also by $q$ values ($q < 0.05$ in Welch's test with post hoc BH procedure). AD-specific nodes, FTLD-specific nodes, and AD-FTLD common nodes are indicated with red, blue, and orange colors.

truncated normal distribution (Supplementary Figure 4) and that commonness of 46.7% was extremely rare (mean +44.6 SD), indicating that the commonness between FTLD core network and AD core network was unexpectedly high and inevitable.

The topological reasons such as the high detectability of proteins by mass analysis could be the reason for such a rare possibility. However, the result of the permutation test with the number of detected proteins clearly denied the possibility and indicated that the extremely high commonness was based on the selection of proteins by pathological changes. Moreover, among 24 phosphoproteome mass analyses (8 mouse models × 3 time points = 24 analyses), 43 FTLD-AD core nodes were detected by >15 analyses. Meanwhile, 1014 nodes were detected by >15 analyses among 24 analyses, indicating that 43 nodes were selected from 1014 nodes based on the FTLD-AD common pathology.

To confirm the validity of our method, we used phosphoproteome data from APP$^{KM670/671NL}$-KI mice[45] as external hold-out data, and examined whether AD core nodes and edges were properly predicted by our method (Supplementary Figure 5a, Supplementary Data 4). The results predicted from four transgenic AD mouse models (5xFAD, APP-Tg, PS1-Tg, PS2-Tg) matched well with the external hold-out data of APP$^{KM670/671NL}$-KI mice (Supplementary Figure 5b), indicating 79.2% of core nodes and 59.8% of core edges were precisely selected (Supplementary Figure 5b). In addition, we expanded the analysis with external hold-out data to the cross-validation analysis of five sets of comparison between one AD model and the other four AD models among APP$^{KM670/671NL}$-KI, 5xFAD-Tg, APP-Tg, PS1-Tg, and PS2-Tg mice (Supplementary Figure 5a). High consistencies (mean precision of core nodes: 81.8%, mean precision of core edges: 64.6%) revealed by the cross-validation analyses further supported the robustness of our analysis (Supplementary Figure 5c).

Moreover, to test the homogeneity of three data sets for each mouse model, we performed cross-validation analyses comparing one data set with the other two data sets (Supplementary Figure 6). Mean absolute error and root mean square error are almost the same among three sets of cross-validation (Supplementary Figure 6).

Big data for steps 1–6 from phosphoproteome analyses are available at (URL: http://suppl.atgc.info/2026/).

**FTLD and AD core pathological networks share limited signals.** The common core network across FTLDs and ADs was further investigated by multiple group comparison for core nodes and by physical interaction records in PPI databases for core edges (Fig. 2a). First, 87.1% of core nodes possessed $q$ values <0.05 in Welch's test with post hoc BH procedure, supporting the validity of the AD-FTLD core network. Second, physical interactions of edges in AD-FTLD core network were reconfirmed by MINT, which is a PPI database constructed based on physical interaction and was employed for generating core network in this study. MINT based on UniProt ID is composed of 26 interaction types, and 25 interaction types directly indicate physical

interaction of edges except the "colocalization" interaction type. The 25 interaction types of MINT supported physical interaction in 13 out of 79 edges of the AD-FTLD core network (13/79 = 16.5%). This ratio was exceptionally high in comparison to the expected ratio of physical interaction by MINT (62,724/20,395 × 20,395=0.015%) calculated from 62,724 edges in MINT reflecting physical interaction and 20,395 human proteins in UniProt used in MINT ($p = 4.60 \times 10^{-35}$ in Fisher's exact test). Another database STRING (ver11.0), which was not used in our study for generating AD-FTLD core network, supported physical interaction in 15 out of 79 edges of the AD-FTLD core network, when "binding" and "reaction" categories in STRING were employed for the test. Considering that physical interactions were experimentally tested in only a part among all edges to generate MINT or STRING, the exceptionally high ratio supported that a rather large part of this core network reflects physical interactions.

The finally determined AD-FTLD core network included a limited number of kinases, microtubule-associated proteins, 14-3-3 proteins, and synapse-related proteins (Fig. 2a, Supplementary Figure 7, 8). Repositioning of the proteins and restructuring the common core network to the protein functions revealed the meaning of the common core network clearly (Fig. 2b). First, the common core network included four Protein Kinase C (PKCalpha, delta, gamma, epsilon) and two Ca$^{2+}$/calmodulin-dependent protein kinase II (CAMK2 alpha, delta) (Fig. 2b). PKCε was connected to b-Raf further supporting activation of the downstream pathway of HMGB1-TLR4 axis (Fig. 2b), which was suggested activated in FTLD and AD pathologies by previous studies[39,41]. CAMK2α and GSK3β were connected with Tau (Fig. 2b) supporting the importance of Tau protein phosphorylation across FTLDs and ADs, irrespective of whether Aβ, TDP43, or Tau is aggregated in neurons or brains. MAP1a, MAP1b, MAP2, and Tau (MAPT) were included in the common core network (Fig. 2b). STX1a (Syntaxin 1a), SYT1 (Synaptotagmin 1), AMPH (Amphysin), DPYSL2 (Dihydropyrimidinase-related protein 2), CRMP1 (Collapsin Response Mediator Protein 1), SYN1 (Synapsin 1), DBN1 (Drebrin 1), RTN4 (Reticulon 4), Dlg2 (Discs Large MAGUK Scaffold Protein 2, PSD-93), Dlg4 (discs large homolog 4, PSD95), SPTBN1 (Spectrin beta chain brain 1) and SPTAN1 (Spectrin alpha non-erythrocymic 1) were all related to synaptic vesicle release, axon guidance, growth cone, neurite outgrowth and/or neuronal migration (Fig. 2b). TRIM28 (Tripartite motif-containing 28) and BIN1 (Bridging Integrator-1/Myc box-dependent-interacting protein 1) were involved in transcription (Fig. 2b). Intriguingly, more than 80% of protein phosphorylation was also detected in human postmortem brains of AD and/or FTLD patients (Supplementary Figure 8).

Among the peripheral proteins of the common core network, only three groups of protein could input the signal from outside of the network even at a relatively loose condition (betweenness > 0), and they were PKC, CAMK2, and KCNMA (potassium calcium-activated channel subfamily M Alpha 1) (Fig. 2b). They were further restricted to PKCα and PKCδ under the strict

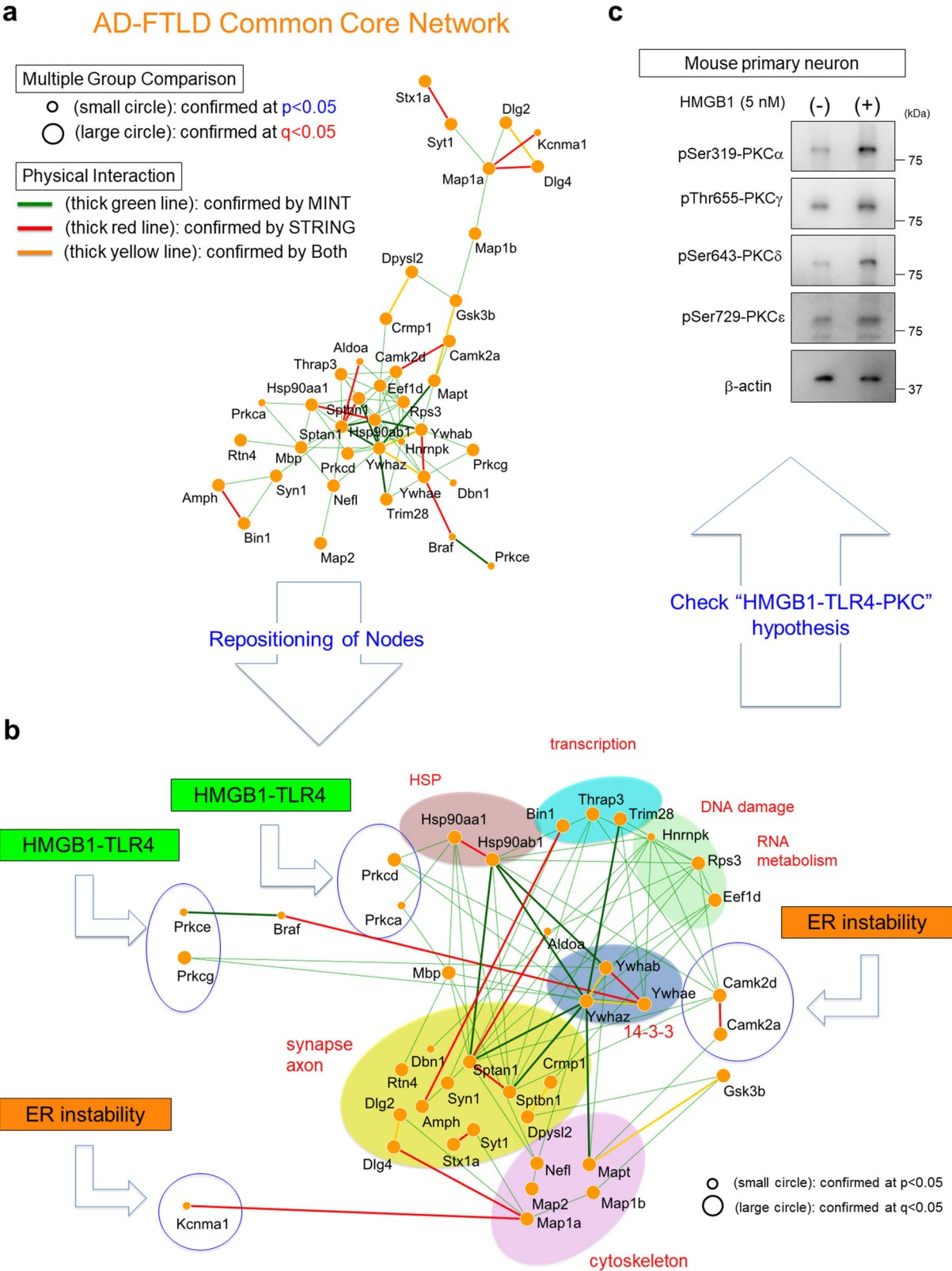

**Fig. 2 Function-based analysis of AD-FTLD common core network. a** AD-FTLD common core network was further examined by multiple group comparison and physical interaction-based PPI. Small circles were selected only by p values ($p < 0.05$ in Welch's test) while large circles were selected also by q values ($q < 0.05$ in Welch's test with post hoc BH procedure). The thick green line indicates an edge whose physical interaction was supported by MINT. The thick red line indicates an edge whose physical interaction was supported by STRING. The thick yellow line indicates an edge whose physical interaction was supported both by MINT and STRING. **b** Next, nodes were repositioned based on protein function and clustered into functional groups. The analysis indicates the gates through which HMGB1-TLR4 or ER-derived $Ca^{2+}$ signals enter the common core network. **c** Western blots of mouse cortical neurons (E15) in primary culture treated with human ds-HMGB1 (HMGBiotech) reveal activation of PKCs. PKCα and PKCδ are activated more prominently.

condition (betweenness >2 SD) (Supplementary Figure 7, 8). The suggested proteins transducing signals into the common core network further supported the significances of HMGB1-TLR4 axis and instability of endoplasmic reticulum (ER) increasing cytoplasmic $Ca^{2+}$, and consistent with our previous results[39,41] (Fig. 2b). HMGB1 is known to be released from necrotic cells but not from apoptotic cells and to induce inflammation in neighboring cells via RAGE or TLR receptors[46,47]. Therefore, we confirmed that extracellular stimuli of HMGB1 to primary cortical neurons actually activate PKCs in the core common network (Fig. 2c).

Moreover, repositioning and restructuring the common core network supported the significant roles of 14-3-3 (YWHA) proteins in transducing pathological signals of FTLD and AD (Fig. 2b, Supplementary Figure 7). 14-3-3 proteins have been implicated in Alzheimer's disease[48–51], Lewy body disease, Parkinson's disease[52–54], and Creutzfeld-Jacob disease[55,56], and our mathematical prediction of the commonality of 14-3-3 proteins across AD and FTLD. In the group, related synapse function, SPTAN1, SPTBN1, and Tau were hubs of the network, and they were phosphorylated at Ser1217/1217(mouse/human), Ser2137/2138(mouse/human), and Ser203/214(mouse/human), respectively (Supplementary Data 1–3). The pathological meaning of site-specific phosphorylation was investigated previously with PGRN$^{R504X}$-KI mice[6].

### Therapeutic effects of human HMGB1 antibody on behavioral phenotypes of FTLD mouse models.

Given that mathematical analysis of phosphoproteome big data supported the significance of transcriptional repression-induced atypical cell death (TRIAD) necrosis with ER instability observed in cortical neurons of FTLD mouse models[39] and the resultant release of HMGB1 causing the secondary damage of surrounding neurons[39,40] for the progression of FTLD diseases, we examined whether inhibition of HMGB1-TLR4 axis by anti-HMGB1 antibody inhibits the progression of FTLD mouse models. We have already generated "human monoclonal antibody against human di-sulfate HMGB1 (hereafter described as an anti-HMGB1 antibody)" as described previously (Tanaka, H. et al. HMGB1 signaling phosphorylates Ku70 and impairs DNA damage repair in Alzheimer's disease pathology. unpublished). The antibody crosses blood–brain barrier (BBB) at the efficiency of 2–8% antibody concentration ratio from plasma to cerebral cortex tissue (Tanaka, H. et al. HMGB1 signaling phosphorylates Ku70 and impairs DNA damage repair in Alzheimer's disease pathology. unpublished). The time points of onset have also been evaluated by Morris water maze test in our previous study[39]. Immediately after the onset of Morris water maze test, we started intravenous injection of anti-HMGB1 antibody once per month (6 μg/injection/mouse = 200 μg/kg BW/injection), which corresponds to 1.2 mg/injection for a patient of 60 kg body weight.

For evaluation of the therapeutic effect by Morris water maze test, we continued for 5 months and performed six times injection (Fig. 3a). In each model at 5 months after the onset, anti-HMGB1 antibody successfully recovered the latency to reach the platform (Fig. 3a). For evaluation of the therapeutic effect by Y-maze test, we continued injection five to six times after the onset of each model determined by Y-maze test (Fig. 3b). In two of four cases (TDP43$^{N267S}$-KI, VCP$^{T262A}$-KI), the therapeutic effect of the anti-HMGB1 antibody appeared immediately after the injection and was sustained while we continued the injection (Fig. 3b). Other models (PGRN$^{R504X}$-KI, CHMP2B$^{Q165X}$-KI) also showed recovery of the symptom within 2 or 3 months after starting the injection (Fig. 3b). For evaluation of the therapeutic effect by fear-conditioning test, we continued injection five to six times

additionally after the onset of each model, and confirmed the therapeutic effect of anti-HMGB1 antibody on TDP43$^{N267S}$-KI, PGRN$^{R504X}$-KI, VCP$^{T262A}$-KI, and CHMP2B$^{Q165X}$-KI mice (Fig. 3c).

It is of note that anti-HMGB1 antibody successfully recovered three memory-linked phenotypes studied here: Morris water maze, Y-maze, and cued fear-conditioning tests because they address different dimensions of memory (long-term, short term) and different neural circuits[57]. Morris water maze test demands recall of long-term memory after days, while Y-maze test demands recall of a short-term memory within a few seconds. Spatial memory examined by Morris water maze and Y-maze tests is stored in the hippocampus[58–60] while emotional memory evaluated by cued fear-conditioning test is stored in the amygdala, prefrontal cortex, and hippocampus[61–63]. The memory could be transferred to and consolidated in the cerebral cortex especially during sleep[64]. Our antibody therapy against HMGB1 was effective on all the factors related to memory, indicating that HMGB1 regulates fundamental pathological elements such as neuronal cell death and/or synaptic function.

Collectively, these results indicated that post-onset administration of anti-HMGB1 antibody ameliorates symptoms of FTLD mouse models irrespective of the mutant gene.

### Therapeutic effects of human HMGB1 antibody on cellular phenotypes of FTLD mouse models.

Next, we investigated pathological changes underlying the therapeutic effect on symptoms. First, TRIAD necrosis detected by pSer46-MARCKS for surrounding neurite degeneration[39,41] and ER marker (KDEL) for ER dilatation was decreased by intravenous injection of anti-HMGB1 antibody (Fig. 4a, b). Second, neuronal DNA damage detected by the markers γH2AX and 53BP1 together with MAP2 was also recovered by administration of anti-HMGB1 antibody (Fig. 4c, d). Third, protein aggregation pathology detected by TDP43 or p62 was partially rescued by anti-HMGB1 antibody (Fig. 4e, f). Co-staining of PSD95 and VAMP2 (Fig. 4g) or of PSD95 and pSer203-Tau (Fig. 4h) confirmed the rescuing effect of anti-HMGB1 antibody on synapse abnormalities as predicted by mathematical analysis of phosphoproteome (Fig. 2).

The results revealed by immunohistochemistry were further supported by western blot analyses (Fig. 4i). Interestingly, PKCα and PKCδ predicted by dynamic network analysis at the severe threshold of core node selection (betweenness > 2 SD) (Fig. 2) were actually changed, while PKCε and PKCγ selected at a relatively loose condition (betweenness > 0) (Supplementary Figure 8) were not changed in phosphorylation actually (Fig. 4i). Tau phosphorylation at Ser203, which was shown linked to synapse instability[6], was also increased in four FTLD mouse models while the phosphorylation was recovered by anti-HMGB1 antibody (Fig. 4i). Consistently, PSD95 was decreased in four FTLD mouse models while the amount was recovered by anti-HMGB1 antibody (Fig. 4i).

Collectively, the therapeutic effect of anti-HMGB1 antibody on various symptoms of FTLD mouse models was considered based on the improvement of secondary neuronal and neurite damages induced by extracellular HMGB1 released from necrotic neurons.

### Analysis of side effects of human HMGB1 antibody.

Finally, we examined the side effects of anti-HMGB1 antibody. We dissected lung, liver, kidney, intestine, spleen, skeletal muscle, heart, and skin from mice after completion of the behavioral test in the protocol (Fig. 3) and used for pathological examinations (Supplementary Figs. 9–16). No obvious inflammation, cell death, or tumor was detected in normal or FTLD model mice receiving

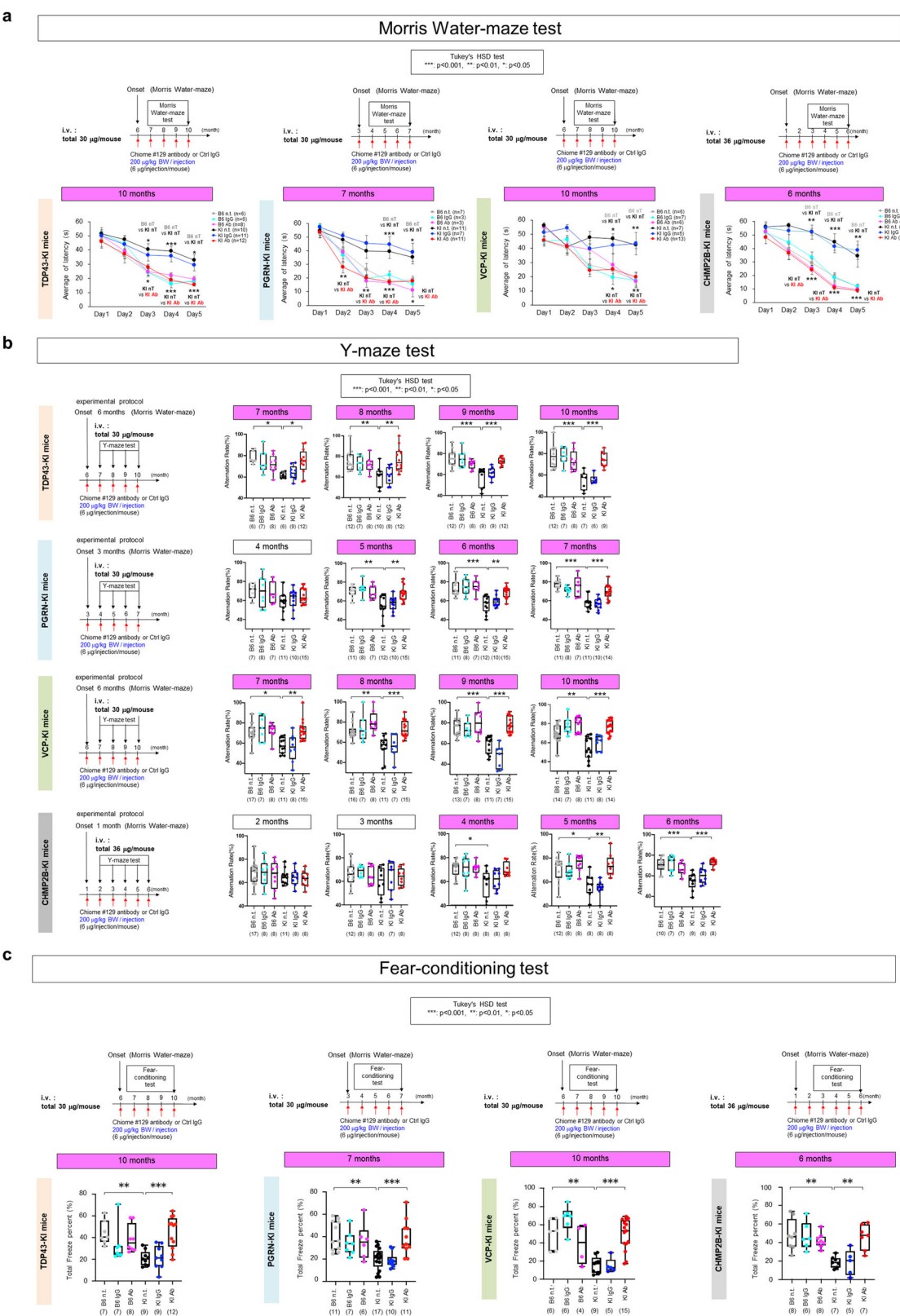

**a** Morris Water-maze test

**b** Y-maze test

**c** Fear-conditioning test

administration of anti-HMGB1 antibody (Supplementary Figs. 9–16).

## Discussion

Our newly developed method for dynamic molecular network analysis by utilizing multi-dimensional vectors and PPI network predicted the core network shifting chronologically during the time course of diseases and shared cross multiple FTLD and AD mouse models. The predicted nodes of the core network were highly plausible and consistent with a number of accumulated experimental and clinical evidence in the field. For instance, a gain of function gene mutations that activate PKCα activity is linked to late-onset familial AD[65], and Aβ oligomer is known to

**Fig. 3 Therapeutic effects of human anti-HMGB1 monoclonal antibody on behavioral symptoms. a** (Upper panel) Experimental protocol of human anti-HMGB1 antibody injection and Morris water maze test for four FTLD model mice were indicated. (Lower graph) Five-day observation of average latency to platform revealed that human anti-HMGB1 antibody significantly recovered the latency. *$p < 0.05$, **$p < 0.01$, ***$p < 0.001$ by Tukey's HSD test. **b** (Left panel) Experimental protocol of human anti-HMGB1 antibody injection and Y-maze test for four FTLD model mice were indicated. (Right graphs) Human anti-HMGB1 antibody significantly recovered the alternation rate (%) at multiple time points. *$p < 0.05$, **$p < 0.01$, ***$p < 0.001$ by Tukey's HSD test. **c** (Upper panel) Experimental protocol of human anti-HMGB1 antibody injection and fear-conditioning test assessment for four FTLD model mice were indicated. (Lower graph) Assessment of total freezing time (%) on day 2 revealed that human anti-HMGB1 antibody significantly recovered the total freezing time. *$p < 0.05$, **$p < 0.01$, ***$p < 0.001$ by Tukey's HSD test. Box plots show median, quartiles, and whiskers, which represent data outside the 25–75th percentile range.

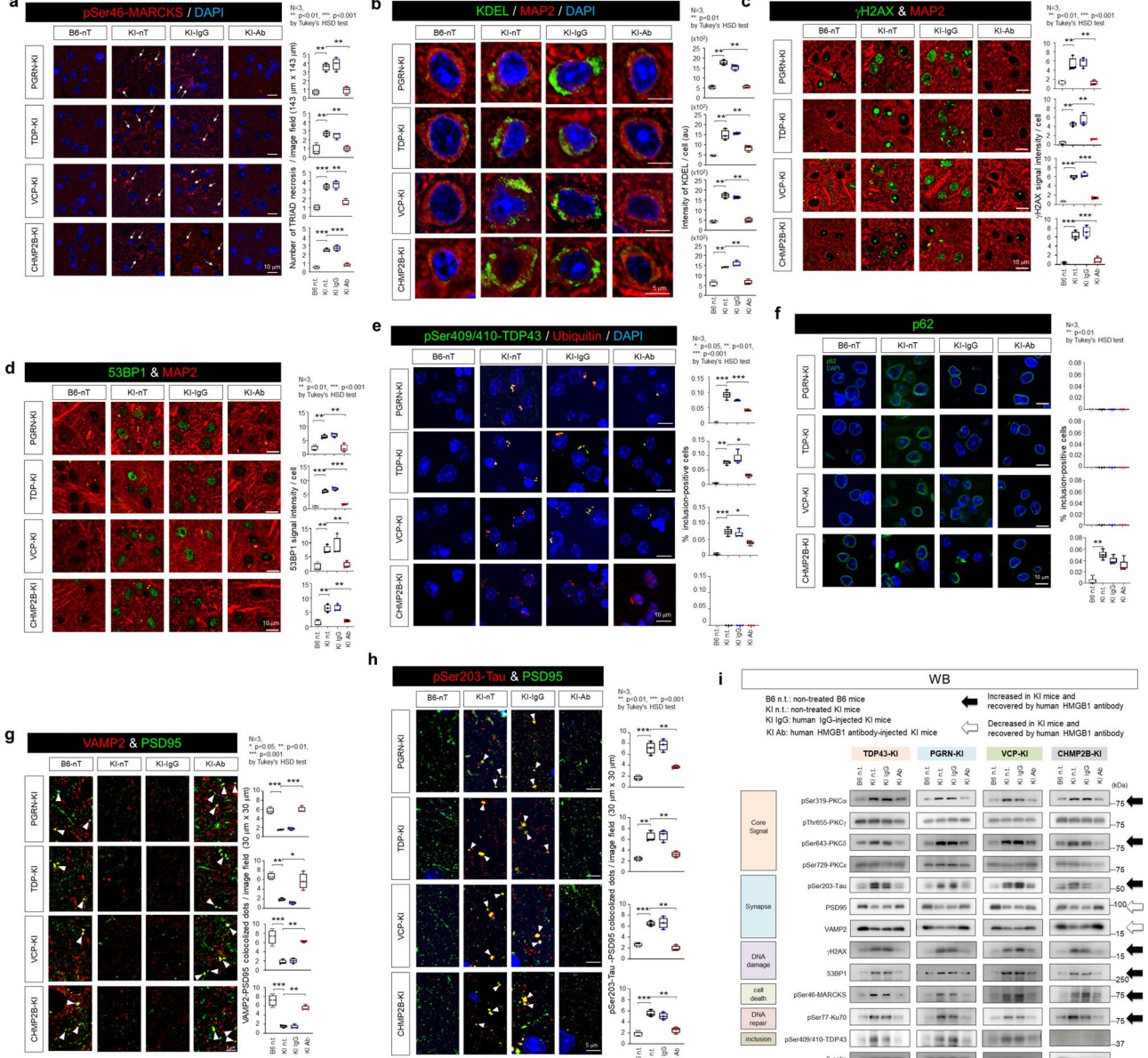

**Fig. 4 Therapeutic effects of human anti-HMGB1 monoclonal antibody.** Neuronal necrosis, neuronal DNA damage, and inclusion body formation were analyzed by immunohistochemistry. **a** pSer46-MARCKS for necrosis, **b** KDEL for necrosis, **c** γH2AX for DNA damage, **d** 53BP1 for DNA damage, **e** pSer409/410-TDP43 for TDP43-positive inclusion body, **f** p62 for p62-positive inclusion body, **g** co-staining of PSD95 and VAMP2 for pre- and post-synapse contact, and **h** co-staining of PSD95 and pSer203-Tau for synapse abnormality. **i** Therapeutic effects of human anti-HMGB1 monoclonal antibody assessed by western blot analysis. Box plots show median, quartiles, and whiskers, which represent data outside the 25–75th percentile range.

affect synapse[66–69]. The consistency of our prediction and previous data support the soundness of our mathematical method. Meanwhile, it is not known whether exactly similar pathologies exist in FTLDs or FTLD spectrum. Various external stimuli, which could be extracellular or intracellular stimuli, activate PKCs or increase intracellular $Ca^{2+}$, while our recent results suggested HMGB1-TLR4 axis in neurons could be the trigger of PKC[39–41].

Therefore, based on the hypothesis predicted by mathematical analysis of phosphoproteome big data, in the FTLD pathology we performed intervention of HMGB1 that might expand neurodegeneration as a mediator of secondary neuronal and neurite damages, and actually confirmed therapeutic effects of anti-HMGB1 antibody on the symptoms and pathologies of four FTLD models possessing mutations in different causative genes. The results strongly suggest anti-HMGB1 antibody as a valuable candidate for developing a disease-modifying therapy (DMT) against FTLDs. The rescuing effect of anti-HMGB1 antibody was also confirmed previously on the symptoms and pathologies of AD model mice[41]. In consistency with the prediction by our dynamic molecular network analysis, the HMGB1-TLR4 axis is shared between ADs and FTLDs. Therefore, anti-HMGB1 antibody might be developed to a DMT for multiple types of dementias including mixed pathologies of Aβ and TDP43 frequently found in sporadic cases of dementia.

Moreover, the consistency between prediction and verification shown in this study indicates that our prediction method is sound and would be useful for analyzing various types of dynamic molecular networks in neurological, psychiatric, and other organ diseases currently intractable and demanding target molecules for radical therapeutics. In addition, since the study was performed by a human monoclonal antibody against human di-sulfate HMGB1, our results could be considered directly for clinical application.

Our mathematical analysis of big data obtained from comprehensive phosphoproteome of multiple AD-FTLD mouse models at multiple time points predicted that a very restricted range of network composed of specific kinases, synapse molecules, cytoskeletal proteins, and 14-3-3 family proteins regulate FTLD and AD pathologies. Though the predicted AD-FTLD common network accords well with a number of previous findings accumulated on synapse dysfunction and cytoskeletal abnormality under neurodegeneration. However, regarding kinases, it is unexpected that only three groups, i.e., four PKCs, two CAMK2s, and one GSK3, are suggested to play key functions in the core pathology. The limited number of kinase groups should be focused on targeting common and/or early-stage pathologies in FTLDs and ADs.

This study has not focused on the specific pathology characterizing each disease or each disease group. However, theoretically, subtraction of AD-FTLD common core network from AD common core network and from FTLD common core network would generate AD-specific and FTLD-specific components in the pathological networks, respectively. Moreover, subtraction of AD common core network from each AD network or subtraction of FTLD common core network from each FTLD network would generate components of the pathological network specific to each disease. Such investigations should be performed in the future to identify AD-specific, FTLD-specific, and disease-specific pathomechanisms.

In conclusion, our new method based on multi-dimensional vector analysis would be applied to analyses of various types of chronologically dynamic molecular networks in this field and in the other fields, and the acquired findings regarding the common core network across multiple dementias would be useful widely for our society.

## Methods

**Generation of four FTLD mouse models.** Generation of PGRN[R504X]-KI mice was reported previously[6]. Knock-in mice harboring the heterozygous PGRN R504X mutation were constructed by insertion of a Neo cassette provided by Unitech (Chiba, Japan) into C57BL/6 J mice. The targeting vector was generated from two constructs. (Construct 1) A PCR product of a 6.5 kbp NotI–XhoI fragment containing the R504X mutation, amplified from a BAC clone (ID: RP23-311P1 or RP23-137J17), was subcloned into vector pBS-DTA (Unitech). (Construct 2) A PCR product of a 3.0 kbp ClaI–XhoI fragment was subcloned into vector pBS-LNL (−) containing the Neo cassette. The BamHI (Blunt)–XhoI fragment from Construct 2 was subcloned between the XhoI (Blunt) and SalI sites of Construct 1, and the resultant plasmid was used as the targeting vector. After linearization with SwaI, the fragment was electroporated into a C57BL/6 J mouse ES clone. Genotyping of ES clones was performed by PCR using the following primers: 5′-CGTGCAATCCATCTTGTTCAAT-3′ (forward), 5′-CATGACCTAACTCAATG CATACCAC-3′ (reverse). Positive clones were subjected to Southern blot analysis. The 5′ and 3′ probes for Southern blots were generated as follows: 5′-probe, 5′-C TGTGTCTCACTAGAAGCATAAGCA-3′ (forward), 5′-ACTAGATTGGGAAAG ACAGTGAATC-3′ (reverse); 3′-probe, 5′- AATTCCAATCCTGTGTGGTCAT AG-3′ (forward), 5′-CCACTTTCTTTCT CCCTTACCCTA-3′ (reverse). Neomycin probes were generated using the following primers: 5′-GAACAAGATGGGATTGC ACGCAGGTTCTCCG-3′ (forward), 5′-GTAGCCAACGGCTATGTCCTGATAG-3′ (reverse). The Neo cassette, present in the F1 mice, was removed by crossing with CAC-Cre mice.

For the construction of targeting vector for VCP[T262A]-KI mouse, PCR product of 5.4 kb NotI–XhoI fragment amplified from Bac clone (ID: RP23-111G9 or RP23-124L1) was subcloned into PspOMI-XhoI of pBS-DTA. Similarly, 2.9 kb BamHI-NotI fragment was amplified from the same Bac clone, subcloned into pBS-LNL(-) that possesses Neo cassette (loxP-Neo-loxP). In all, 4.5 kb fragment (1.6 kb Neo cassette +2.9 kb fragment) was cleaved out by NotI. In all, 0.8 kb XhoI-SmaI fragment with T262A mutation was amplified by PCR from the same Bac clone and digested with XhoI and SmaI. The 0.8 kb and 4.5 kb fragments were subcloned into XhoI-NotI sites of the targeting vector. After linearization of the targeting vector with NotI, the fragment was electroporated into C57BL/6 J background ES cells. VCP[T262A]-KI mice were discriminated against non-transgenic littermate by PCR using primers 5′-ATATGCTCTCACTGTATGGTATTGC-3′ and 5′-TCC AGATGAGCTTAGAAGATTAGAA-3′ that amplify DNA fragment containing LoxP sequence.

For the construction of targeting vector of CHMP2B[Q165X]-KI mouse, PCR product of 5.7 kb ClaI-SalI fragment amplified from Bac clone (ID: RP23-13H5 or RP23-273E18) was subcloned into ClaI-SalI of pBS-DTA (Unitech, Chiba, Japan). 3.0 kb SacII-NotI fragment was amplified from the Bac clone and subcloned into pBS-LNL(+) that possesses Neo cassette (loxP-Neo-loxP). SacII-NotI, and 4.6 kb fragment (1.6 kb Neo cassette +3.0 kb fragment) were subcloned into SacII-ClaI sites of targeting vector. CHMP2B[Q165X]-KI mice were discriminated against non-transgenic littermate by PCR using primers 5′-TGGATTTTATTTATGTCTGAA TGTG-3′ and 5′-ATAAGCAACTTCACAAGGCATCTTA-3′ that amplify DNA fragment containing LoxP sequence.

For the construction of targeting vector of TDP43[N267S]-KI mouse, 5.9 kb ClaI-SalI fragment amplified from Bac clone (ID: RP23-364M1 or RP23-331P21) and subcloned into ClaI-SalI of pBS-TK (Unitech, Chiba, Japan). In all, 2.7 kb SacII-NotI fragment was amplified from the Bac clone and subcloned into pA-LNL(+) that possesses Neo cassette (loxP-Neo-loxP). In all, 4.3 kb fragment (1.6 kb Neo cassette +2.7 kb fragment) was subcloned into SacII-ClaI sites of targeting vector. TDP43[N267S]-KI mice were discriminated from non-transgenic littermate by PCR using primers 5′-ACAGTTGGGTGTGATAGCAGGTACT-3′ and 5′-TCGAGA ATTACAGGAATGTATCATC-3′ that amplify DNA fragment containing LoxP sequence. The detailed method of mouse generation is described in our recent paper[39].

**AD model mice.** 5xFAD transgenic mice overexpressing mutant human APP (770) with the Swedish (KM670/671NL), Florida (I716V), and London (V717I) familial Alzheimer's disease (FAD) mutations, and human PS1 with FAD mutations (M146L and L285V) were purchased from The Jackson Laboratory (34840-JAX, Bar Harbor, ME, USA). Both the APP and PS1 transgenes were under the control of the mouse Thy1 promoter[70]. The backgrounds of the mice were C57BL/SJL, which was produced by crossbreeding C57BL/6J female and SJL/J male mice. APP-KI mice possess a single human APP gene with the Swedish (KM670/671NL), Arctic (E693G), and Beyreuther/Iberian (I716F) mutations[45].

**2D LC-MS/MS analysis.** Adult mice at the age of 1, 3, 6 months were killed, and cerebral cortexes were obtained for 2D LC-MS/MS analysis. Mouse cerebral cortex samples were dissolved in lysis buffer (100 mM Tris-HCl (pH 7.5), 2% SDS (sodium dodecyl sulphate), 1 mM DTT) with protease inhibitor cocktail (#539134, 1:100 dilution, Calbiochem, CA, USA) were incubated at 100°C for 15 min. After centrifugation (12,000 $g$ × 10 min), the supernatants (1.5 mg protein in 200 μl) were added to 100 μl of 1 M triethylammonium (TEAB) (pH 8.5), 3 μl of 10% SDS, and 30 μl of 50 mM tris-2-carboxyethylphosphine. After 1 hour incubation at 60°C, cysteine residues were blocked with 15 μl of 200 mM methyl methanethiosulpho-nate for 10 min at room temperature. Samples were then digested with trypsin

(150 μg) in 80 mM CaCl₂ for 24 hours at 37°C. Phosphopeptides were enriched using the Titansphere Phos-Tio Kit (GL Sciences Inc., Tokyo, Japan) and desalted using Sep-Pak Light C18 catridge column (Waters Corporation, MA, USA). The samples were dried and dissolved with 25 μl of 100 mM TEAB (pH 8.5). The phosphopeptides were then labeled separately using the iTRAQ Reagent multiplex assay kit (AB SCIEX Inc., CA, USA) for 2 hours at room temperature. After the samples were mixed together, the aliquots were dried and then re-dissolved in 1 mL of 0.1% formic acid.

The labeled phosphopeptide samples were subjected to strong cation exchange chromatography using TSK gel SP-2SW column (TOSOH, Tokyo, Japan) on Prominence UFLC system (Shimadzu, Kyoto, Japan). The flow rate was 1.0 mL/min with solution A (10 mM KH₂PO₄ (pH 3.0), 25% acetonitrile). Elution was performed with solution B (10 mM KH₂PO₄ (pH 3.0), 25% acetonitrile, 1 M KCl) in a gradient ranging from 0 to 50%. The elution fractions were dried and dissolved in 100 μL of 0.1% formic acid.

Each fraction was analyzed using a DiNa Nano-Flow LC system (KYA Technologies Corporation, Tokyo, Japan) at the flow rate was 300 nL/min. For the Nano-LC, samples were loaded onto a 0.1 mm × 100 mm C18 column with solution C (2% acetonitrile and 0.1% formic acid) and eluted with a gradient of 0–50% solution D (80% acetonitrile and 0.1% formic acid). The ion spray voltage to apply sample from Nano-LC to Triple TOF 5600 System (AB SCIEX, Framingham, MA, USA) was set at 2.3 kV. The Information-Dependent Acquisition setting was 400–1250 m/z with two to five charges. Analyst TF software (version 1.5, AB SCIEX, Framingham, MA, USA) was used to identify each peptide. The quantification of each peptide was based on the TOF-MS electric current detected during the LC-separated peptide peak, adjusted to the charge/peptide ratio. The signals were analyzed by Analyst TF and processed by Protein Pilot software (version 4, AB SCIEX, Framingham, MA, USA) as described in the following.

**Mass data analysis**. Mass spectrum data of peptides were acquired and analyzed by Analyst TF (version 1.5, AB SCIEX, MA, USA). Using the results we retrieved corresponding proteins from the public database of mouse protein sequences (UniProtKB/Swiss-Prot, downloaded from http://www.uniprot.org on 22 June 2010) by Protein Pilot (version 4, AB SCIEX, Framingham, MA, USA A) that employs Paragon algorithm. Tolerance for the search of peptides by Protein Pilot was set to 0.05 Da for the MS and 0.10 Da for the MS/MS analyses, respectively. "Phosphorylation emphasis" was set at sample description, and "biological modifications" were set at processing specification of Protein Pilot. The confidence score was used to evaluate the quality of identified peptides, and the deduced proteins were grouped by Pro Group algorithm (AB SCIEX, Framingham, MA, USA) to exclude redundancy. The threshold for protein detection was set at 95% confidence in Protein Pilot, and proteins with more than 95% confidence were accepted as identified proteins.

Quantification of peptides was performed through analysis of iTRAQ reporter groups in MS/MS spectrum that are generated upon fragmentation in the mass spectrometer. In quantification of peptides, bias correction that assumes the total signal amount of each iTRAQ should be equal was employed to normalize signals of different iTRAQ reporters. After bias correction, the ratio between reporter signals in KI mice and that of control mice (peptide ratio) was calculated.

These results in peptide ratios were imported to Excel files from summaries of Protein Pilot for further data analyses. The quantity of a phosphopeptide fragment was calculated as the geometric mean of signal intensities of multiple MS/MS fragments including the phosphorylation site. The difference between the KI mouse group and the control group was judged significant when the *p* value was <0.05 (*N* = 3) by Welch's test (Step 1).

**Molecular network analyses**. Static molecular network analyses were performed as described previously[5,6]. To generate a pathological PPI network based on the changed phosphopeptides in FTLD model mice, a list of proteins from significantly changed phosphopeptides was created. UniProt accession numbers were added to the proteins in the list. Proteins whose UniProt accession numbers were not listed in the Human Genome Project (GNP) (https://cell-innovation.nig.ac.jp/GNP/index_e.html) database were removed from the list of proteins. The selected proteins were used for the generation of the pathological PPI network of the FTLD model mouse based on the integrated database of GNP including BIND (http://www.bind.ca/), BioGrid (http://www.thebiogrid.org/), HPRD (http://www.hprd.org/), IntAct (http://www.ebi.ac.uk/intact/site/index.jsf), and MINT (https://mint.bio.uniroma2.it/). One additional edge and node were added to the selected nodes on the PPI database. A database of GNP-collected information was created on the Supercomputer System available at the Human Genome Center of the University of Tokyo.

For analysis of PPI-based network structure, the betweenness centrality score was used as the degree of importance of proteins in the PPI-based network. A node with a high betweenness score can be thought of as an important hub node that is connected with many nodes in terms of the information flow of paths. A node was determined as "high" at mean ± 3 SD in each group of the model mice. R/igraph 1.0.0 package was employed to calculate betweenness centrality scores.

To choose the core modules of the pathological network, edges in the network were also evaluated in terms of chronological changes of phosphopeptides. Each phosphorylated site was characterized by a 3D vector using average amounts of

phosphopeptides including them through three time points (1 M, 3 M, and 6 M). Correlation values of two changed phosphorylated sites were calculated as cosine of two vectors from the phosphorylated sites. An edge was determined as 'highly correlated' at *r* ≥ 0.9, where *r* was an absolute value of correlation.

Finally, the common core network across FTLDs and ADs was constructed from the selected nodes and edges. Nodes with "high" score of betweenness in ≥2FTLDs and ≥2ADs were selected as the core nodes. "Highly correlated" edges in ≥2FTLDs and ≥2ADs were also selected as the core edges. Only nodes that were connected by core edges were used as the common core network across FTLDs and ADs.

**Examinations of mathematical approach with external hold-out and cross-validation**. To verify the estimated common core network among multiple models, a hold-out analysis was employed using additional three APP-KI model mice at three time points (1, 3, or 6 months). The additional APP-KI model mice were analyzed independently as an external hold-out of the existing four AD models (5xFAD, APP-Tg, PS1-Tg, and PS2-Tg). Core networks estimated from the four ADs and that from the hold-out were compared in terms of the number of shared nodes or edges. Fivefold cross-validation using those five AD models was performed to assess the robustness of the proposed approach. In each trial, four of the five models were assigned as an estimation data set, and the rest of one model was separated as validation data. Two networks were created from estimation data set or validation data, and precision of node selection was calculated as a ratio of shared nodes between two networks to the total nodes selected from validation data.

**Evaluation of random variation among samples**. A threefold cross-validation analysis of error prediction for the estimated correlation values was performed to evaluate random variations among samples. For each AD or FTLD model at every time point, three mice were separated randomly into two groups to compare the estimated correlation values. In each data set, the mean absolute error (MAE) and root mean square error (RMSE) of the estimated correlations were calculated. MAE and RMSE were calculated as follows:

$$MAE_{e,g1,g2} = \frac{\sum_{i=1}^{N} |corr_{e,g2} - corr_{e,g1}|}{N} \qquad (1)$$

$$RMSE_{e,g1,g2} = \sqrt{\frac{\sum_{i=1}^{N} \left| corr_{e,g2} - corr_{e,g1} \right|^2}{N}} \qquad (2)$$

where *e* denotes an edge, *g1*, *g2* denote group 1 and group 2 in the data set, *N* denotes the number of possible edges in the data set, and *corr_{e,g}* is a correlation value of an edge *e* in an estimated network of group *g*.

**Permutation-based simulation to evaluate the accuracy of node selection**. To approximate the power of the proposed mathematical approach to estimate the network, a permutation-based simulation of node selection was performed. In all, 10,000 bootstrap samples were created by choosing 73 nodes as AD core nodes and 62 as FTLD core nodes randomly from the 1965 proteins detected by phospho-proteome mass analyses in eight model mice. %commonness of nodes was calculated for every bootstrap sample and the distribution of %commonness was evaluated. Mean and SD of %commonness were estimated under the assumption of a truncated normal distribution.

**Preparation of mouse primary cortical neuron**. Mouse primary cortical neurons were obtained from C57BL/6J mouse embryo (E15). Cerebral cortexes were minced into fine pieces and incubated with 0.05% trypsin at 37 °C for 10–15 min and then with DNase at a final concentration of 25 μg/mL for another 5 min at 37 °C. Dulbecco's Modified Eagle Medium (DMEM) (Gibco, NY, USA) containing 50% FBS was added to the dissociated cells, and cells were centrifuged at 100 × g for 3 min. resuspended in 10 mL of 10% FBS/DMEM, gently triturated with blue tips, and filtered through a nylon mesh (Flacon 2350, BD Biosciences, NJ, USA). The isolated cells were centrifuged in a Neurobasal medium and placed at 6 × 10⁵ cells in six well-plate (Corning, Glendale, AZ, USA) coated with polyethyleimine (Sigma-Aldrich, MO, USA). Seventy-two hours after plating, arabinosyl cytosine (Sigma-Aldrich, St. Louis, MO, USA) was added to the culture medium (0.5 μM) to prevent unnecessary glial cell growth.

Seven days after seeding, disulfide HMGB1 was used at 5 nM (HM-120, HMGBiotech, Milano, Italy) for 3 h. Cells were collected and homogenized with a plastic homogenizer (Bio-Masher II, Nippi, Tokyo, Japan) following the addition of lysis buffer [100 mM Tris-HCl (pH 7.5, Sigma, MO, USA), 2% SDS (Sigma, St. Louis, MO, USA), 1 mM DTT (Sigma, St. Louis, MO, USA), and a protease inhibitor cocktail (Calbiochem, #539134, 1:200 dilution)]. The lysates were boiled at 100 °C for 15 min. After centrifugation (16,000 g × 10 min at 4 °C), the supernatants were diluted with an equal volume of sample buffer [125 mM Tris-HCl (pH 6.8, Sigma, St. Louis, MO, USA), 4% SDS (Sigma, St. Louis, MO, USA), 20% glycerol (Wako, Osaka, Japan), 12% mercaptoethanol (Wako, Osaka, Japan), and 0.005% BPB (Nacalai, Kyoto, Japan)]. Sodium dodecyl

sulphate–polyacrylamide gel electrophoresis (SDS-PAGE) and western blot analysis were then performed.

**Anti-Human HMGB1 antibody preparation.** The Autonomously Diversifying Library (ADLib®) system[71,72] was used to isolate HMGB1-specific human mono-clonal antibodies. The human ADLib® library (Chiome Bioscience Inc., Tokyo, Japan), which was constructed by replacing the chicken IgM heavy and light chain loci in DT40 cells with the corresponding human IgG1 gene sequences, was screened for the binding capability of membrane-bound IgG1 against biotinylated HMGB1 protein. Disulfide HMGB1 (HMGBiotech S.r.l., Milano, Italy) protein was used as an antigen. The sequences of the variable regions of the immunoglobulin genes from the isolated HMGB1-specific clones were determined from genomic DNAs. Recombinant anti-HMGB1 IgG1 antibodies were transiently expressed by transfecting FreeStyle™ 293-F cells (Thermo Fisher Scientific, Waltham, MA, USA) with pFUSE-CHIg-hG1 (InvivoGen, San Diego, CA, USA) and pFUSE2-CLIg-hk (InvivoGen, San Diego, CA, USA) vectors carrying sequences encoding the variable regions of the heavy and light chains of HMGB1-specific clones, respectively, which were then purified via protein A chromatography. A detailed method of antibody generation is described in our recent paper (Tanaka, H. et al. HMGB1 signaling phosphorylates Ku70 and impairs DNA damage repair in Alzheimer's disease pathology. unpublished).

**Injection of human anti-HMGB1 monoclonal antibody.** Four FTLD model mice or their background mice (C57BL/6 J) received intravenous injections of 6 μg/30 g body weight control IgG (human IgG isotype control, #12000 C, Thermo Fisher Scientific, Waltham, MA, USA) or the anti-HMGB1 monoclonal antibody (Chiome, final volume of 100 μL) once a month into the tail vein following the experimental schedule (Fig. 3).

**Behavioral test.** Three types of behavioral tests were performed for male mice as described previously[5,73] at indicated age in the Fig. or legend. In the Morris water maze test, mice performed four trials (60 sec) per day for 5 days, and the latency to reach the platform was measured. Y-shape maze consisting of three identical arms with equal angles between each arm (O'HARA & Co., Ltd, Tokyo, Japan). Mice were placed at the end of one arm and allowed to move freely through the maze during an 8 min session. The percentage of spontaneous alterations (indicated as an alteration score) was calculated by dividing the number of entries into a new arm that was different from the previous one with the total number of transfers from an arm to another arm. In the fear-conditioning test, the freezing response was measured 24 h after the conditioning trial (65 dB white noise, 30 s + foot shock, 0.4 mA, 2 s) in the same chamber in the absence of a foot shock.

**Immunohistochemistry.** Five μm thickness sections were de-paraffinized in xylene, rehydrated, dipped in 0.01 M citrate buffer (pH 6.0), and microwaved at 120℃ for 15 minutes. After permeabilization by 0.5% triton-X 100 containing phosphate-buffered saline (PBS), sections were incubated with blocking solution (10% FBS containing PBS) for 60 min at room temperature. Then, sections were incubated with primary antibody for 12 hours at 4℃, washed with PBS three times at room temperature, incubated with secondary antibodies at room temperature for 1 hour. All the procedures were performed in parallel for the mouse groups compared.

The antibodies used for immunohistochemistry were dilution as following, mouse anti-γH2AX antibody at 1:200 (Ser139, #05-636, Millipore, Burlington, MA, USA); rabbit anti-53BP1 at 1:5000 (NB100-304, Novus Biologicals, Centennial, CO, USA); rabbit anti-phospho TDP43 (Ser409/410) antibody at 1:5000 (TIP-PTD-P02, Cosmo Bio Co. Ltd., Tokyo, Japan); mouse anti-ubiquitin antibody at 1:10,000 (P4D1, cell signaling technology, Danvers, MA, USA); mouse anti-p62 antibody at 1:200 (#610497, BD bioscience, San Jose, CA, USA); rabbit anti-phospho-MARCKS (Ser46) at 1:2000 (GL Biochem [Shanghai] Ltd., Shanghai, China); mouse anti-KDEL antibody at 1:200 (ADI-SPA-827, Enzo Life Science, Farmingdale, NY, USA); rabbit anti-MAP2 antibody at 1:1000 (ab32454, Abcam, Cambridge, UK); mouse anti-MAP2 antibody at 1:50 (sc-32791, Santa Cruz Biotechnology, Dallas, TX, USA); rabbit anti-PSD95 antibody at 1:200 (D74D3, Cell Signaling Technology, Danvers, MA, USA); mouse anti-VAMP2 antibody at 1:200 (GTX634812, GeneTex, Inc., Alton Pkwy Irvine, CA, USA); rabbit anti-phospho Tau (Ser203(mouse)/Ser214(human)) antibody at 1:200 (ab170892, Abcam, Cambridge, UK); mouse anti-PSD95 antibody at 1:200 (MA1-045, Thermo Fisher Scientific, Waltham, MA, USA); donkey anti-rabbit IgG Cy3-conjugated at 1:500 (711-165-152, Jackson Laboratory, Bar Harbor, ME, USA); donkey anti-mouse IgG Alexa488 at 1:1000 (A21202, Molecular Probes, Eugene, OR, USA); donkey anti-mouse IgG Cy3-conjugated at 1:500 (715-165-150, Jackson Laboratory, Bar Harbor, ME, USA); donkey anti-rabbit IgG Alexa488 at 1:1000 (A21206, Molecular Probes, Eugene, OR, USA). All images were acquired using confocal microscopy (Olympus FV1200IX83, Tokyo, Japan).

**Hematoxylin–Eosin staining.** Dissected organ tissues were fixed with 4% paraformaldehyde for 24 hours, washed with PBS, and embedded in paraffin. In all, 5 μm thickness sections were made using a microtome (Yamato Kohki Industrial Co., Ltd., Saitama, Japan). The sections were de-paraffinized by xylene, rehydrated

washed with water, and stained with Carrazzi's Hematoxylin solution (1.15938.0025, Merck, Darmstadt, Germany) for 10 min at room temperature. After washing with tap water for 10 min, sections were stained with eosin solution for 5 min at room temperature (0.25% eosin diluted with 80% ethanol, 051-06515, Wako, Osaka, Japan) and de-hydrated with ethanol. Sections were immersed with xylene for clearing and covered. Images were obtained by light microscopy (BX53, Olympus, Tokyo, Japan).

**Western blot.** Mouse cortical tissues were dissolved in lysis buffer (100 mM Tris-HCl (pH 7.5), 2% SDS) with protease inhibitor cocktail (#539134, 1:100 dilution, Calbiochem, CA, USA) for 1 hour at 4℃. After centrifugation (12,000 g x 10 min), the supernatants were added to the equal volume of sample buffer (125 mM Tris-HCl pH 6.8, 4% (w/v) SDS, 12% (v/v) 2-mercaptoethanole, 20% (v/v) glycerol, and 0.005% (w/v) bromophenol blue). BCA method (Pierce BCA Protein Assay Kit; Thermo Scientific, Waltham, MA, USA) was used to equalize the protein amounts in samples. Samples were separated by SDS-PAGE, trans-ferred onto polyvinylidene difluoride membrane Immobilon-P (Millipore) by semi-dry method, blocked with 5% milk in TBST (10 mM Tris/HCl, pH 8.0, 150 mM NaCl, 0.05% Tween20), and reacted with primary and secondary antibodies diluted in TBST with 0.1% skim milk or Can Get Signal solution (Toyobo, Osaka, Japan) as follows: rabbit anti-phospho PKCα (Ser319) antibody at 1:10,000 (AP0560, NeoScientific, Cambridge, MA, USA); rabbit anti-phospho PKCγ (Thr655) antibody at 1:10,000 (ab5796, Abcam, Cambridge, UK); rabbit anti-phospho PKCδ (Ser643) antibody at 1:3000 (#9376, Cell signaling tech-nology, Danvers, MA, USA); rabbit anti-phospho PKCε (Ser729) antibody at 1:3000 (ab63387, Abcam, Cambridge, UK); rabbit anti-phospho Tau (Ser203 (mouse)/Ser214(human)) at 1:10,000 (ab170892, Abcam, Cambridge, UK); rabbit anti-PSD95 antibody at 1:5000 (D74D3, Cell Signaling Technology, Danvers, MA, USA); mouse anti-VAMP2 antibody at 1:5000 (GTX634812, GeneTex, Inc., Alton Pkwy Irvine, CA, USA); mouse anti-γH2AX antibody at 1:2000 (Ser139, #05-636, Burlington, MA, USA); rabbit anti-53BP1 antibody at 1:50,000 (NB100-304, Novus Biologicals, Centennial, CO, USA); rabbit anti-phospho TDP43 (Ser409/410) antibody at 1:6000 (TIP-PTD-P02, Cosmo Bio Co. Ltd., Tokyo, Japan); rabbit anti-phospho-MARCKS (Ser46) at 1:100,000 (GL Biochem [Shanghai] Ltd., Shanghai, China); rabbit anti-phospho-Ku70 (Ser77/ Ser78) antibody at 1:100,000, ordered from Cosmo Bio Co., Ltd., Tokyo, Japan); horseradish peroxidase (HRP)-linked anti-rabbit IgG, 1:3000 (NA934, GE healthcare, Chicago, IL, USA); HRP-linked anti-mouse IgG, 1:3000 (NA931, GE healthcare, Chicago, IL, USA). Incubation with primary or secondary antibodies was performed for 12 hours at 4℃ or for 1 h at room temperature, respectively. ECL Prime Western Blotting Detection Reagent (RPN2232, GE healthcare, Chicago, IL, USA) and luminescent image analyzer (ImageQuant LAS 500, GE healthcare, Chicago, IL, USA) were used to detect proteins.

**Statistics and reproducibility.** Statistical analysis for biological examinations was performed using R (version 3.6.2), GraphPad prism (version 8), or Microsoft Excel for Microsoft 365. Two group comparison was performed using unpaired Student's t test for normal distributions. For multiple groups comparisons, Tukey's HSD test was employed to calculate p values under the assumption of normally distributed variables. All data are represented in Box-Whisker plots with the original data points shown as dots.

**Ethics.** This study was performed in strict accordance with the recommendations in the Guide for the Care and Use of Laboratory Animals of Japanese Government and of the National Institutes of Health. All the experiments were approved by the Committees on Gene Recombination Experiments, Human Ethics, and Animal Experiments of the Tokyo Medical and Dental University (G2018-082C, 2011-22-3/O2020-002, and A2019-218C2).

**Reporting summary.** Further information on research design is available in the Nature Research Reporting Summary linked to this article.

## Data availability

The mass spectrometry proteomics data of four AD mouse models (5xFAD, APP-Tg, PS1-Tg, PS2-Tg) have been deposited to the ProteomeXchange Consortium via the PRIDE partner repository with the data set identifier PXD001292[5], and the proteomics data of four FTLD models and APP$^{KM670/671NL}$-KI mice have been deposited to the ProteomeXchange with the data set identifier PXD027119. Results of comprehensive phosphoproteome analyses and PPI-based molecular network analyses for Fig. 1 are available on the author's Website (http://suppl.atgc.info/2026/). All other data generated or analyzed during this study are included in this article and its supplementary information files.

## Code availability

The custom scripts for analyses and evaluations are available at https://github.com/hmmhdnr/molecular_network_analysis_multidimentional_vector and the DOI-minting repository Zenodo at https://doi.org/10.5281/zenodo.5073892[74].

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

## Acknowledgements
This work was supported by the Strategic Research Program for Brain Sciences (SRPBS), Brain Mapping by Integrated Neurotechnologies for Disease Studies from Japan agency for Medical research and Development (Brain/MINDS) from Japan agency for Medical research and Development (AMED), a Grant-in-Aid for Scientific Research on Innovative Areas (Foundation of Synapse and Neurocircuit Pathology, 22110001/ 22110002) from the Ministry of Education, Culture, Sports, Science and Technology of Japan (MEXT), a Grant-in-Aid for Scientific Research A (16H02655, 19H01042) from Japan Society for the Promotion of Science (JSPS). Human brain samples were obtained from Brain Bank for Aging Research in Tokyo Metropolitan Geriatric Hospital and Institute of Gerontology and Fukushimura Brain Bank which is supported by JSPS KAKENHI Grant Number JP 16H06277 (CoBiA) and AMED under Grant Number JP21wm0425019. We thank professor Satoru Miyano (TMDU) and professor Seiya Imoto (The University of Tokyo) for kind collaboration in the usage of supercomputer at The University of Tokyo, and professor Takaomi Saido (RIKEN), professor Takashi Saito (Nagoya City University), and RIKEN BioResource Center for providing APP-KI mice.

## Author contributions
Meihua Jin, Xiaocen Jin, Kyota Fujita, Hikari Tanaka, and Kazuhiko Tagawa performed experiments, analyzed data, and wrote the paper. Shigeo Murayama and Hiroyasu Akatsu prepared human samples, analyzed human data, and wrote the paper. Hidenori Homma designed and performed mathematical experiments, and wrote the paper. Hitoshi Okazawa designed this project, co-designed experiments, and wrote the paper.

## Competing interests
The authors declare no competing interests.
