## [Peer Review File · Communications Biology]

Reviewers' comments:

Reviewer #1 (Remarks to the Author):

1) General comments:

The authors present a dynamic molecular network analysis method of phosphoproteomics data to identify a shared core pathological protein-protein interaction sub-network between Alzheimer's disease (AD) and frontotemporal lobar degeneration (FTLD), using both data from mouse models and human postmortem brains. The network model is used to predict a therapeutic intervention approach, involving the inhibition of the protein HMGB1, and the authors report that the experimental testing of this intervention in four FTLD mouse ameliorated neuropathology and symptoms.

Overall, the manuscript is clearly written and the proposed analysis approach is logical. The fact that the authors mention in lines 112-113 that an anti-HMGB1 antibody has already been shown previously to ameliorate pathological progression and symptoms in AD model mice limits the novelty of the proposed intervention, even though the transfer of the approach from AD to FTLD mouse models represents a new aspect of this work. My main concerns are however that the authors do not present a validation of the mathematical approach using either a cross-validation or a validation on external holdout data to test the robustness and replicability of the network analysis results, the distribution assumptions behind the statistical tests and correlation analyses are not checked, the public interaction data used for the protein-protein interaction network construction is not filtered by confidence, and no adjustments of the p-value significance scores for multiple hypothesis testing have been mentioned (which is particularly concerning given the limited sample size, see Major comments below).

2) Specific comments for revision:

a) Major comments

1.) More information should be provided in the introduction on the rationale for why the authors expect common network alterations in FTLD and AD. The introduction only mentions that both diseases are dementias, but does not provide prior background information on why shared network alterations are expected in the specific brain tissue studied, and why the mouse models are considered as comparable.

2.) The authors suggest on page 4, lines 124-125 that the anti-HMGB1 therapeutic effect in FTLD mouse models verifies the soundness of their new mathematical model. However, while the experimental findings are important, they only provide anecdotal evidence for the utility and correctness of the mathematical model, and a cross-validation or an external validation on independent hold-out data is required to confirm that the findings are robust and can be replicated independently.

3.) Page 5, lines 137-138: The number of replicate samples used for the Welch Test analyses is very small ($N=3$, see line 138). Moreover, it is not clear whether the Welch Test p-values have been adjusted for multiple hypothesis testing, instead the authors only mention a significance cut-off of 0.05 for the nominal, unadjusted p-value. Have the authors checked that the normal distribution assumption for the Welch Test is fulfilled? Both a normality test should be shown and the nominal p-values should be adjusted for multiple hypothesis testing.

4.) For the definition of core edges in the correlation network construction, the authors specify a correlation threshold of 0.9. Does this refer to the Pearson correlation, and if so, have the authors checked that the data is normally distributed (otherwise, using the Spearman or distance correlation would be more appropriate)? Moreover, the limited number of data points for the correlation analysis raises the question how robust the results are with respect to random variation in the data. A cross-validation analysis with standard deviations for the estimated correlations should be conducted to

verify the robustness of the correlation scores in this setting.

In line 156, the authors mention that 50% of the core nodes and core edges are shared between the AD and FTLN network, but is this significantly larger than expected by chance (the topological criteria alone may already result in very similar networks, even without taking the phosphoproteomics data into account)? A statistical test or a permutation-based simulation should be shown to verify that the overlap between the networks is significantly larger than expected by chance.

5) On page 15, lines 462-466, the authors mention the databases used to construct the PPI network; however, no filtering criteria are discussed. Since the databases contain many false-positive interactions derived from experimental methods to assess indirect interactions (e.g. tandem affinity purification), a filtering of interactions with low confidence should be applied to ensure the quality of the network, or alternatively, the authors should discuss how they prevent false conclusions associated with false-positive interactions during the network analysis stage.

b) Minor comments

1.) Page 3, line 70: Replace "preciously" with "previously"

2.) Page 3: The reference to the EBI training course website, which is not available anymore, should be replaced by an up-to-date and relevant reference.

3.) Page 3, line 87: Replace "are recently called as " by "have been referred to since recently as "

4.) Page 4, line 121-122: Replace "we confirmed therapeutic effect" by "we confirmed a therapeutic effect"

5.) In general, the manuscript contained various spelling and grammatical errors, and I recommend a revision of the language with the help of a native English speaker.

Reviewer #2 (Remarks to the Author):

Prediction and verification of the FTLN-AD common pathomechanism based on dynamic molecular network analysis

Multiple gene mutations have been identified in familial frontotemporal lobar degeneration (FTLN) while no single gene mutations exists in sporadic FTLN. In order to identify a target in spite of this complexity, the authors used a novel systems biology approach based on phosphoproteomics approach from three novel knockin mouse transgenic models of FTLN and three transgenic models of familial forms of Alzheimer's Disease (AD). From this approach, authors selected a common core molecule High mobility group 1 (HMGB1) protein. HMGB1 is both a nuclear factor and a secreted protein. Furthermore, HMGB1 is a potent mediator of inflammation. Manipulation of HMGB1 levels with an antibody against the protein allows the rescue of both cellular phenotypes and behavioral phenotypes.

This study is based on sound results. Rescue of both cellular and behavioral phenotypes is a main advance in the field.

Some points need to be improved and clarified.

Mathematical analysis of sequential phosphoproteome big data

Other submitted papers by the same group indicate their use of the APP KI mice from Saito-Saido

group. Did the authors analyze the phosphoproteome changes as compared to other models. KI models are smart as there will be no overexpression artifacts in contrast to transgenic mice with random integration of the transgene.
This point needs to be discussed.

FTLD and AD core pathological networks share limited signals

Function of HMGB1 needs to be presented in this paragraph, in particular its role in inflammation (Scaffidi P, Misteli T, Bianchi ME. Release of chromatin protein HMGB1 by necrotic cells triggers inflammation. Nature. 2002 Jul 11;418(6894):191-5.).

Therapeutic effects of human HMGB1 antibody on symptoms of FTLD mouse model

It could be better to indicate

Therapeutic effects of human HMGB1 antibody on behavioral phenotypes of FTLD mouse models.

Three memory-linked phenotypes have been studied: Morris water maze, Y maze and cued fear conditioning. They address different dimensions of memory (long-term, short term) and different circuits (see for instance Josselyn and Tonegawa, Science, 2020).

This information needs to be summarized here.

Therapeutic effects of human HMGB1 antibody on pathologies of FTLD mouse model

I suggest

Therapeutic effects of human HMGB1 antibody on cellular phenotypes of FTLD mouse models

Discussion

Rescue approaches are not presented in the discussion. They need to be discussed in this part.

Minor corrections:

Line 37. Multiple gene mutations cause familial frontotemporal lobar degeneration (FTLD) instead of Multiple gene mutations causes familial frontotemporal lobar degeneration (FTLD)

Line 297 SNPs that activate PKC α activity are linked to late-onset familial AD 48

Mutations reported in ref 48 are gain of function mutations and not SNP, for Late-Onset Alzheimer's Disease (LOAD).

>>> We thank reviewers for kind and constructive comments.

Reviewers' comments:

Reviewer #1 (Remarks to the Author):

1) General comments:

The authors present a dynamic molecular network analysis method of phosphoproteomics data to identify a shared core pathological protein-protein interaction sub-network between Alzheimer's disease (AD) and frontotemporal lobar degeneration (FTLD), using both data from mouse models and human postmortem brains. The network model is used to predict a therapeutic intervention approach, involving the inhibition of the protein HMGB1, and the authors report that the experimental testing of this intervention in four FTLD mouse ameliorated neuropathology and symptoms.

Overall, the manuscript is clearly written and the proposed analysis approach is logical. The fact that the authors mention in lines 112-113 that an anti-HMGB1 antibody has already been shown previously to ameliorate pathological progression and symptoms in AD model mice limits the novelty of the proposed intervention, even though the transfer of the approach from AD to FTLD mouse models represents a new aspect of this work.

>>> We thank reviewer #1 very much for very kind evaluation.

My main concerns are however that the authors do not present a validation of the mathematical approach using either a cross-validation or a validation on external holdout data to test the robustness and replicability of the network analysis results,

>>> We performed additional analysis of proteome data from APP-KI mice as external holdout data, and compared the results from previously used data from four mouse models (5xFAD, APP-Tg, PS1-Tg, PS2-Tg) (new Supplementary Figure 5). We also extended the comparison to all combinations using one of them as an external holdout data, and performed cross-validation analyses in

additional four sets of comparison among five AD models (new Supplementary Figure 5).

>>> Moreover, to test the homogeneity of three datasets for each mouse model, we performed cross validation analyses comparing one dataset with the other two datasets (new Supplementary Figure 6). As shown, mean absolute error and root mean square error are almost same among three sets of cross validation (new Supplementary Figure 6).

the distribution assumptions behind the statistical tests and correlation analyses are not checked, the public interaction data used for the protein-protein interaction network construction is not filtered by confidence, and no adjustments of the p-value significance scores for multiple hypothesis testing have been mentioned (which is particularly concerning given the limited sample size, see Major comments below).

>>> We really appreciate these constructive comments and advices from reviewer #1. We performed requested analyses as described in the following.

2) Specific comments for revision:

a) Major comments

1.) More information should be provided in the introduction on the rationale for why the authors expect common network alterations in FTLD and AD. The introduction only mentions that both diseases are dementias, but does not provide prior background information on why shared network alterations are expected in the specific brain tissue studied, and why the mouse models are considered as comparable.

>>> We improved Introduction by adding new paragraphs to discuss the requested issues; rational for AD-FTLD common network, usage of cerebral cortex samples, and usage of mouse as main sample.

2.) The authors suggest on page 4, lines 124-125 that the anti-HMGB1 therapeutic effect in FTLD mouse models verifies the soundness of their new mathematical model. However, while the experimental findings are important, they only provide anecdotal evidence for the utility and correctness of the mathematical model, and a cross-validation or an external validation on independent hold-out data is required to confirm that the findings are robust and can be replicated independently.

>>> Following the advice of the reviewer, we used data from APP-KI mice as external holdout data, and examined our mathematical method (new Supplementary Figure 5). In addition, we performed the similar analyses in five comparisons of one mouse model and the other four mouse models (new Supplementary Figure 5). High precision ratios of core nodes and core edges in all comparisons supported the validity of our mathematical method.

Further to evaluate quality of N=3 datasets, we performed cross validation and confirmed that mean absolute error and root mean square error are almost same among three sets of cross validation (new Supplementary Figure 6).

3.) Page 5, lines 137-138: The number of replicate samples used for the Welch Test analyses is very small (N=3, see line 138). Moreover, it is not clear whether the Welch Test p-values have been adjusted for multiple hypothesis testing, instead the authors only mention a significance cut-off of 0.05 for the nominal, unadjusted p-value. Have the authors checked that the normal distribution assumption for the Welch Test is fulfilled? Both a normality test should be shown and the nominal p-values should be adjusted for multiple hypothesis testing.

>>> We added new Supplementary Figure 1 to test whether our data (log values) are distributed normally. As shown in Supplementary Figure 1b and 1c, we tested the normality test by employing quantile-quantile plot. The log values in our analysis show normal distributions with heavy tails, which is often observed in log values, indicate that our values basically compose normal distributions.

>>> We used Benjamini-Hochberg (BH) procedure for multiple hypothesis testing, and the results of selected nodes and edges by BH are integrated into networks in Figure 1 and 2.

4.) For the definition of core edges in the correlation network construction, the authors specify a correlation threshold of 0.9. Does this refer to the Pearson correlation, and if so, have the authors checked that the data is normally distributed (otherwise, using the Spearman or distance correlation would be more appropriate)?

>>> We neither used the Pearson correlation nor the Spearman correlation. We employed cosine metrics for which normal or non-normal distribution is essential.

Moreover, the limited number of data points for the correlation analysis raises the question how robust the results are with respect to random variation in the data. A cross-validation analysis with standard deviations for the estimated correlations should be conducted to verify the robustness of the correlation scores in this setting.

>>> We performed the requested cross-validation analysis with the original mouse data sets, and investigated the robustness of the correlation scores by calculating mean absolute error and root mean square error (Supplementary Figure 6).

In line 156, the authors mention that 50% of the core nodes and core edges are shared between the AD and FTLD network, but is this significantly larger than expected by chance (the topological criteria alone may already result in very similar networks, even without taking the phosphoproteomics data into account)? A statistical test or a permutation-based simulation should be shown to verify that the overlap between the networks is significantly larger than expected by chance.

>>> Following the advice of the reviewer we examined by permutation test how rarely we could expect 46.7% of commonness. By bootstrap procedure, we randomly selected 73 nodes (equal to the number of AD core nodes) and 62

nodes (equal to the number of FTLN core nodes) from 1,965 phosphoproteins detected by phosphoproteome in total, repeated the selection 10,000 times, and tested their overlap. The result of calculations revealed that %commonness revealed a truncated normal distribution (Supplementary Figure 4a) and that commonness of 46.7% was extremely rare (mean + 44.6 SD), indicating that the commonness between FTLN core network and AD core network was unexpectedly high and inevitable.

>>> The reviewer asked us *“the topological criteria alone may already result in very similar networks, even without taking the phosphoproteomics data into account”*? Our approach was the topological selection of disease-biased datasets. However, permutation test with proteins detected by our phosphoproteome analysis instead of proteins changed in pathological conditions (proteins whose phosphorylation was changed in disease model mice), which is topological selection of non-disease-biased datasets, did not support this hypothesis.

5) On page 15, lines 462-466, the authors mention the databases used to construct the PPI network; however, no filtering criteria are discussed. Since the databases contain many false-positive interactions derived from experimental methods to assess indirect interactions (e.g. tandem affinity purification), a filtering of interactions with low confidence should be applied to ensure the quality of the network, or alternatively, the authors should discuss how they prevent false conclusions associated with false-positive interactions during the network analysis stage.

>>> As the reviewer suggested, PPI databases include various types of interaction including indirect or false positive interactions. We used MINT database, which was used for our screening and based on physical interaction, and STRING ver11.0, which was not used for our screening, and examined the ratio of edge whose physical interaction was supported, among edges composing the AD-FTLN network. The ratio was exceptionally high (confirmed by Fisher’s exact test). Considering not all combinations of all proteins were

examined to make MINT, the exceptionally high ratio supported that a rather large part of this core network reflects physical interactions.

b) Minor comments

1.) Page 3, line 70: Replace "preciously" with "previously"

>>> We corrected the error.

2.) Page 3: The reference to the EBI training course website, which is not available anymore, should be replaced by an up-to-date and relevant reference.

>>> We corrected the error of URL address. We also apologize some critical papers had not been referred here, which we now added.

3.) Page 3, line 87: Replace "are recently called as " by "have been referred to since recently as "

>>> We followed the advice of the reviewer, and corrected the phrase.

4.) Page 4, line 121-122: Replace "we confirmed therapeutic effect" by "we confirmed a therapeutic effect"

>>> We corrected the error.

5.) In general, the manuscript contained various spelling and grammatical errors, and I recommend a revision of the language with the help of a native English speaker.

>>> We had asked a professional editor to correct spelling and grammatical errors.

Reviewer #2 (Remarks to the Author):

Prediction and verification of the FTLD-AD common pathomechanism based on dynamic molecular network analysis

Multiple gene mutations have been identified in familial frontotemporal lobar degeneration (FTLD) while no single gene mutations exists in sporadic FTLD. In order to identify a target in spite of this complexity, the authors used a novel systems biology approach based on phosphoproteomics approach from three novel knockin mouse transgenic models of FTLD and three transgenic models of familial forms of Alzheimer's Disease (AD). From this approach, authors selected a common core molecule High mobility group 1 (HMGB1) protein. HMGB1 is both a nuclear factor and a secreted protein. Furthermore, HMGB1 is a potent mediator of inflammation. Manipulation of HMGB1 levels with an antibody against the protein allows the rescue of both cellular phenotypes and behavioral phenotypes.

This study is based on sound results. Rescue of both cellular and behavioral phenotypes is a main advance in the field.

>>> We thank reviewer #2 very much for very kind evaluation.

Some points need to be improved and clarified.

Mathematical analysis of sequential phosphoproteome big data

Other submitted papers by the same group indicate their use of the APP KI mice from Saito-Saido group. Did the authors analyze the phosphoproteome changes as compared to other models. KI models are smart as there will be no overexpression artifacts in contrast to transgenic mice with eandom integration of the transgene.

This point needs to be discussed.

>>> We performed phosphoproteome analyses of APP-KI mice as external data and revealed the similar molecular network to those of other AD mouse models, supporting the validity of our method and result.

FTLD and AD core pathological networks share limited signals

Function of HMGB1 needs to be presented in this paragraph, in particular its role in inflammation (Scaffidi P, Misteli T, Bianchi ME. Release of chromatin protein

HMGB1 by necrotic cells triggers inflammation. Nature. 2002 Jul 11;418(6894):191-5.).

>>> We added some sentences about HMGB1 release from necrotic cells and inflammation in neighboring cells by referring the paper.

Therapeutic effects of human HMGB1 antibody on symptoms of FTLD mouse model

It could be better to indicate

Therapeutic effects of human HMGB1 antibody on behavioral phenotypes of FTLD mouse models.

>>> We changed the title of section following the advice of the reviewer.

Three memory-linked phenotypes have been studied: Morris water maze, Y maze and cued fear conditioning. They address different dimensions of memory (long-term, short term) and different circuits (see for instance Josselyn and Tonegawa, Science, 2020).

This information needs to be summarized here.

>>> Following the advice of the reviewer, we added a paragraph summarizing types of memory and relevant neural circuit, together with the meaning of behavioral improvement by anti-HMGB1 antibody.

Therapeutic effects of human HMGB1 antibody on pathologies of FTLD mouse model

I suggest

Therapeutic effects of human HMGB1 antibody on cellular phenotypes of FTLD mouse models

>>> We changed the title of section following the advice of the reviewer.

Discussion

Rescue approaches are not presented in the discussion. They need to be discussed in this part.

>>> In previous version of MS, we discussed a rescue effect of HMGB1 antibody on FTLD models. We added some sentences on the effect on AD models and strengthened the discussion.

Minor corrections:

Line 37. Multiple gene mutations cause familial frontotemporal lobar degeneration (FTLD) instead of Multiple gene mutations causes familial frontotemporal lobar degeneration (FTLD)

>>> We corrected the grammatical error.

Line 297 SNPs that activate PKC α activity are linked to late-onset familial AD 48 Mutations reported in ref 48 are gain of function mutations and not SNP, for Late-Onset Alzheimer's Disease (LOAD).

>>> We corrected the improper usage of "SNPs" here.

** See the Nature Portfolio author and referees' website at www.nature.com/authors for information about policies, services and author benefits

Communications Biology is committed to improving transparency in authorship. As part of our efforts in this direction, we are now requesting that all authors identified as 'corresponding author' create and link their Open Researcher and Contributor Identifier (ORCID) with their account on the Manuscript Tracking System prior to acceptance. ORCID helps the scientific community achieve unambiguous attribution of all scholarly contributions. You can create and link

your ORCID from the home page of the Manuscript Tracking System by clicking on 'Modify my Springer Nature account' and following the instructions in the link below. Please also inform all co-authors that they can add their ORCID to their accounts and that they must do so prior to acceptance.

If you experience problems in linking your ORCID, please contact the Platform Support Helpdesk.

Our flexible approach during the COVID-19 pandemic

If you need more time at any stage of the peer-review process, please do let us know. While our systems will continue to remind you of the original timelines, we aim to be as flexible as possible during the current pandemic.

COMMSBIO - This email has been sent through the Springer Nature Tracking System NY-610A-NPG&MTS

Confidentiality Statement: □□ This e-mail is confidential and subject to copyright. Any unauthorised use or disclosure of its contents is prohibited. If you have received this email in error please notify our Manuscript Tracking System Helpdesk team at <http://platformsupport.nature.com> .

Details of the confidentiality and pre-publicity policy may be found here <http://www.nature.com/authors/policies/confidentiality.html>

Privacy Policy | Update Profile

REVIEWERS' COMMENTS:

Reviewer #1 (Remarks to the Author):

The authors have addressed the main comments and their revisions have improved the manuscript.